# Circular Economy Assessment in Recycling of LLDPE Bags According to European Resolution, Thermal and Structural Characterization

**DOI:** 10.3390/polym14040754

**Published:** 2022-02-15

**Authors:** Ricardo Ballestar, Celia Pradas, Fernando Carrillo-Navarrete, Javier Cañavate, Xavier Colom

**Affiliations:** 1Research Department of Sphere Group Spain, Av. Miguel Servet s/n, 50180 Zaragoza, Spain; rballestar@sphere-spain.es (R.B.); cpradas@sphere-spain.es (C.P.); 2Chemical Engineering Department, Universitat Politècnica de Catalunya BarcelonaTECH. ESEIAAT, Colom 1, 08222 Terrassa, Spain; fernando.carrillo@upc.edu (F.C.-N.); francisco.javier.canavate@upc.edu (J.C.)

**Keywords:** mechanical recycling, polyethylene degradation, circular economy, recycled LLDPE, thermal analysis

## Abstract

According to the Circular Economy Package promoted by the European directive, plastic bags companies must use in their formulations a percentage of polyethylene waste (industrial and/or domestic) greater than 70%. Following that regulation requires an understanding of its consequences in the final product from an industrial point of view. This manuscript analyzes the thermal and morphological changes related to the tear resistance of linear-low density polyethylene (LLDPE) samples from industrial waste generated by the company Sphere Spain subjected to the degradation produced by the recycling cycles. The process is analogue to the industrial, starts from samples in pellets then a film by blow extrusion is obtained (odd steps) and posteriorly this film is recycled to pellets again (even steps). The results obtained show that the LLDPE samples develop two crystalline structures (CS1 and CS2) which evolve differently through the recycling cycles with a tendency to decrease in crystallinity due to degradation that is not the same for the process of obtaining film or recycling to pellet. The molecules with a more linear structure and a longer chain break and branch. The more branched structure increases and tends to crosslinking. This leads to a decrease in tear strength in the longitudinal direction, which is not so evident in the transversal direction. The samples could admit four recycling cycles with and acceptable tear resistance. The longitudinal tear strength value decreases by 40% for each film and 20% in the case of tearing in the transverse direction. The results obtained in this research work show that the regulations included in the cited circular economy package can be applied in the manufacture of consumer bags, helping also to reduce the dependence of manufacturers on fluctuations in delivery by collapses in shipping.

## 1. Introduction

The evolution to a system based on a circular economy with minimum waste, has arisen the interest of the chemical industry. The recycling and disposal of plastics, due to their massive production, is one of the main concerns in the path towards a cleaner system of production.

According to circular economy principles, a system should tend to restoration or regeneration. In the case of plastics materials this is interpreted as the possibility of recycling plastic materials and the recovery of energy. In order to improve their circularity, plastics have a big potential. Nowadays, their recycling percentage is still low, even though in recent years there has been an important increase [1].

Legislative resolutions promoted by the EU will encourage recycling, especially considering the ambitious objectives of the recent Circular Economy Package and its action plans, which set as common goal to attain a 65% in the recycling of packaging waste by 2025 and a 75% in 2030. These plans are documented in the European Commission’s Circular Economy package [2] and in the New Plastics economy reported by Ellen Mac Arthur Foundation [3].

The plastics industry is aware that the use of recycled material can help to reduce costs while decreasing environmental impact. Replacing virgin material with recycled thermoplastic is a good practice in processes such as injection or extrusion.

The low percentage of plastic recycling is partially a consequence of some difficulties external to the plastic processing itself. Most of the plastic waste is generated in households, mainly packaging, which is usually composed of a heterogeneous mixture of plastics. This entails a need of collection, separation, and grinding stages, making the process more complex. The presence of organic and inorganic pollutants in these post-consumer plastic materials require also washing and purification prior to their use [4,5].

Another important feature and limitation of the plastic recycling is the thermomechanical degradation caused during reprocessing. The processing of a plastic causes the breaking of the polymeric chain, the introduction of functional groups as consequence of the oxidation, increased branching and crosslinking of the chains [6]. These phenomena affect to the properties of the material, reducing its crystallinity, hardness and elongation at break. Some interesting studies have focused in the changes that take place when a polymer (especially polyethylene) is reprocessed. Jin et al. [7] observe the scission of chains and crosslinking, Oblak et al. [8], consider additionally the branching of the polymer and Cruz and Zanin [9] point out the oxidation.

At an industrial level, the validation of a process involving recycled material from different sources to obtain a product in accordance with the relevant regulations, becomes a major challenge. The processing parameters need to be adjusted in a specific way in order to obtain the desired properties when using recycled plastics as an input. This is undoubtedly a challenge, named post-industrial recycling, which is not yet solved and it is object of industrial and academic research.

Polyethylene is a commodity polymer. A plastic material present in many sectors, especially in the field of packaging, where its main uses include bags, containers and others representing the 39.9% of Europe production [10]. Its wide use, particularly as low-density polyethylene, is related to its good qualities and versatility, including low cost, good processability and environmental resistance. Then, in order to follow the European regulations and achieve a higher percentage of recycling in the field of packaging, the study of the reprocessing of the low-density polyethylene is essential. An insight in the possibilities of reintroducing the polyethylene waste in the fabrication process and the effect on the properties of the final product is crucial if this practice has to be applied industrially effect of using.

The aim of this work is just related to providing an understanding of the consequences of introducing polyethylene waste in a real industrial product. The effect caused by the consecutive recycling of linear low-density polyethylene (LLDPE) after being successively incorporated into film extrusion industrial processes for the manufacture of bags is studied. The variation of the physical, thermal and mechanical properties, due to the changes in the structure and morphology of the chains are evaluated. The main objective is to analyze specifically the structural and thermal changes produced in the LLDPE during the recycling in a real industrial process, that take place in Sphere Group Spain SL.

## 2. Materials and Methods

### 2.1. Materials and Samples Processing

Linear low-density polyethylene LLDPE (Dow 2645.01 G) supplied by Dow Chemical Iberica (Tarragona, Spain). The material has a density of 0.918 g/cm^3^ and an MFI of 0.85 g/10 min (at 190 °C/2.16 Kg) according to ASTM D792 and ASTM D1238 respectively [11]. This grade contains a primary antioxidants phenol type and secondary compounds mainly based on phosphites that extend the useful life of the material and prevent its degradation.

The samples obtained in order to evaluate the effect of the degradation depart from neat LLDPE and are submitted to blow extrusion obtaining a film followed by a recovery process that results again in pellets. This process, from pellets to film and again production of pellets has been performed several times, giving a total of 12 processing steps which are described in Figure 1.

The nomenclature of the analysis samples is based on the number of thermo-oxidative steps that the polyethylene has been processed. Therefore, it has the following coding: step zero corresponds to the original LLDPE pellets. The sequence of processes continues with: the odd number of steps that refer to the films obtained through the blown extrusion process and the even number steps that represent the pellets obtained through the recycling process.

First extrusion is made from the neat LLDPE, thus obtaining the first film that corresponds to step one (Step 1). This collected film resulting from the extrusion process is recycled in the recovery machine, to obtaining the second step pellets (Step 2). These collected pellets is used in the extrusion corresponding to the next step and so on.

### 2.2. Extrusion and Blowing to Obtain the Film

Figure 2 shows the pictures of the equipment used to prepare the samples. The extrusion processes were carried out in a Kiefel Kirion 80/70 F Bilayer coextruder No. 713220 (Freilassing, Germany) PE in the form of pellets is fed through the hopper to the extruder and enters through the feeding zone. Once inside, the material melts and mixes through pressure and friction along the screw. This coextruder is constituted by two different screws A: 80/26 F Kirion Tipo HEM 2407 mm and B: 70/26 F Kirion HEM 2082 mm.

For the extrusion of the material 85 mm × 148 mm M25-40-100 filters have been used. Temperature control zones from the feeding zone to the outlet through the die. 170/180/190/190/190–195/195/200/200/200/205/205 (°C).

All extrusion processes were carried out at a speed of 250 Kg/h, maintaining the polymer temperature of 210 ± 5 °C and a pressure of 530 ± 20 bar (A) and 460 ± 30 bar (B). At the outlet of the die, the material is blown and collected in reels with a width of 147 cm and a thickness of 30 microns.

### 2.3. Recycling to Obtain the Pellets

The recycling of the films obtained after each extrusion was carried out in an Erema 1514 TVE PLUS No. P10/155 10128179 recycler (Ansfelden, Austria).

The film is introduced into the recover machine. In the first zone or feeding zone, the material is cut, mixed and crushed with an average feeding speed of 700 Kg/h. Once the film is divided in small fragments, it passes to the screw where the material is again mixed and melt with a temperature profile of: 135/190/210/210/220/210/200/200/200/240 (°C). Finally, the melt leaves the granulator in the form of pellets. This material has been designed as recycled pellets.

### 2.4. Calorimetric Analysis by DSC

The calorimetric analysis was performed using a Mettler DSC1 calorimeter, (Zurich, Switzerland) calibrated using indium (heat flow calibration and temperature calibration) and zinc (temperature calibration) standards. Samples of approximately 10 mg of the mass were deposited in 40 µL aluminum pans in air atmosphere to test performance. The sample is heated at a rate of 10 °C/min to 180 °C to erase the thermal history of the processing. After that, it is cooled to 30 °C and left for 5 min for stabilization, after that it is heated again to 270 °C. To ensure complete oxidation, the sample is left 20 min at 270 °C as shown in Figure 3. DSC was used to determine crystallinity, crystallization and melting temperatures (Tc, Tm) and oxidation temperature (T_OX_).

The percentage of crystallinity of LLDPE (%*χ_c_*) is determined by the obtained melting enthalpy (Δ*H_m_*) and the theoretical enthalpy corresponding to 100% pure polyethylene ($\Delta H_{m}^{0}$ = 293 J/g) according to Equation (1) [12]


(1)
$$\%\chi_{c}=\frac{\Delta H_{m}}{\Delta H_{m}^{0}}\;\;$$


### 2.5. Structural Characterization by FTIR

Chemical structure of LLDPE degraded samples was determined using FTIR analysis performed by means of a Nicolet iS10 spectrometer from Thermo Scientific (Waltham, MA, USA). The device had an ATR attachment with a diamond crystal. 

To determine crystallinity, the bands relation suggested by Zerbi et al. [13] was used; where, the spectral bands of doublets 1472–1462 cm^−1^ corresponding to flexural vibrations: 1472 cm^−1^ (amorphous phase) and 1462 cm^−1^ (crystalline phase) are used. Spectra were registered at 2 cm^–1^ resolution and 40 scans in the range of 1400–1500 cm^−1^, in which the typical LLDPE signals related to -CH_2_- deformation can be observed.

### 2.6. Tear Resistance Test

The Elmendorf tear properties of all blown films were measured, using a IDM Elmendorf DEA-80 Tear tester (San Sebastian, Spain). Following the procedures described in ISO 6383-2, two film sections of 76 mm × 63 mm are cut with a sample cutter from each test film produced and their thickness measured. The 76 mm length of each section was used as the direction of tear. A 20 mm slit was made at the center of the edge perpendicular to the direction being tested.

## 3. Results and Discussion

### 3.1. Calorimetric Analysis

DSC shows a bimodal graph (Figure 3) where a doublet is generally observed in the crystallization and melting zone. As presented in previous research, this doublet is related to two different crystalline structures present in the material [14].

Craig and White have studied polyethylene crystalline structures and their changes under degradation [15]. Al-Salem et al. provided information about the kinetics of degradation and its correlation with TGA results as presented in this article [16]. Zhang et al. [17] on their part, have described the existence of several crystalline structures in some types of polyethylene in blown films.

For polyethylene blown film, two types of structures have been proposed, the row-nucleated and the α-axis. In the row-nucleated structure model, due to the influence of the comonomer with long-chain branched polyethylene two main types of crystals can be formed depending on the stress applied during the processing of the material. The difference between the two types of crystallizations is related to the type of growth of the lamellas radially [18].

According to Zhang et al., in the case of LLDPE, the row-nucleated morphology is not always achieved, and the morphology displays spherulite-like superstructure and random lamellar arrangements. The longer polyethylene sequences show controllable and homogeneous amount of short branches along the chains. This allows the chains to stack and fold in a way that is organized forming a spherulite superstructure with a low orientation [19].

This behavior is found in blown film. In this case, the DSC samples are heated and then observed during the cooling process and the conditions will be different from those reported by Zhang, but the possibility of finding different types of crystalline structures in LLDPE is still valid and is confirmed by the bimodal graph obtained in DSC.

The results show a structure, that is named type I (CS1), which could be related to one of the regions with a greater number of branches in the main chain of polyethylene. These branches generate a lower capacity to form interchain interactions and therefore the associated melting temperature of this regions is slightly lower, (Tm1) around 112 °C. 

The other observed structure, type II (CS2) could correspond to a more linear main chain area, with fewer branches, or in any case represents a structure, which allows a greater capacity for interaction between polymeric chains and therefore shows a higher melting temperature (Tm2) of approximately 122 °C [20].

During the processing steps carried out in this study, the thermal characterization presented multiple changes, which show how these structures are affected after the successive extrusion and recycling processes of the material.

#### 3.1.1. Crystallization

The DSC thermograms (Figure 4) show the cooling zone of the sample, where crystallization parameters of the material can be studied.

Initially, in the first samples, a single crystallization peak is observed around 105 °C, very intense and narrow, which means that crystallization take place at a high rate. The crystallization in samples corresponding at Steps 0 and 1, takes place in a unique way.

Progressively and special from the third step, a widening of the peak and a shoulder appears at the DSC graph. This means that two different phenomena, two types of crystallization are taking place at slightly different temperatures, two crystallizations are overlapping and the relative amount of each varies when increasing the number of steps. When the number of steps increase and especially after step six, the peak appearing around 100 °C is more prominent and wider which means that the structure of the crystal presents less homogeneity.

The widening of the crystallization peak when the number of steps increases is also a sign of the degradation of the samples. The re-processing induces degradation, causing chain breakdown, generating imperfect crystalline structures [21].

A small increase in the crystallization temperature is observed when the processing steps increase, indicating an increase of the onset temperature related to relatively smaller amounts of smaller crystals, which can achieve an ordered structure faster, are formed. Development of interchain crosslinking could also appear due to the creation of radicals during the process of degradation keeping chains in an ordered structure.

From third to eighth steps, the two peaks corresponding to different crystal structures named CS1 and CS2 can be observed. Initially there is a higher percentage of type II crystalline structure (CS2) and with the progress of the steps, CS2 is decreasing and increasing the percentage of the structure type I (CS1). This means that processing affects to the polymer breaking the main chains and generating radicals that result in more branching, causing a relative increase of the structure CS1 vs. CS2 [22].

From step 9, the clear differentiation between the two crystalline structures disappears. One unified single signal is observed, at a lower temperature, corresponding to structure type I (CS1). This is representative of the degradation of the material, the breaking of the chains and the increase in ramifications [23,24].

Table 1 shows the progress of crystallization temperatures of each sample (recycled pellet and film) as a function of the number of processing steps. The crystallization temperatures related to structure CS2 increase, both in pellet and film samples. On the contrast, crystallization temperature of structure CS1 tends to decrease slowly or remains constant. According to the previous discussion, as the samples degrade, the fraction of CS2 decreases, the part that disappears is the less stable part and the remaining crystals are the ones with higher temperature of crystallization. Instead, the fraction of CS1 increases leading to a structure that will include more defects and tend to have lower crystallization temperature.

#### 3.1.2. Melting Temperature

Figure 5 shows DSC thermograms corresponding to the melting temperature [25]. The structure type I (CS1) is observed in the peak of lower intensity with a melting temperature of 112.5 °C, the structure type II (CS2) appears at higher temperature (121 °C) with greater intensity and definition.

This differentiation of peaks is maintained from step 0 to step 8. Among these first eight steps, different phenomena are observed. First of all the difference in the melting temperature of both structures. In the first steps, the temperature difference between both peaks is 10 °C (this is better appreciated in Table 1). As the number of steps increases, the melting temperature of the CS1 increases to 116 °C. This phenomenon is related to a relative improvement of the packing in the more branched chains. When degradation takes place, there are more chains available to form structures type I (CS1) and the free radicals produced during degradation are able to link together these chains, forming some slightly crosslinked and more compacted structures, which may help to increase the melting point of these areas [26] 

On the contrary, the CS2 structures present a melting temperature of 122 °C in the first steps but as they undergo the processing steps, their melting temperature tends to remain constant or even slightly decrease. This is due to degradation, the long and less branched chains become fragmented, creating more imperfections in the crystals and lowering the melting temperature. 

As previously commented, Table 1 provides a clearer image of the evolution of the temperatures of both crystalline structures.

The second important trend shown in Figure 5 is related to the intensity of the peaks. The peak corresponding to structure CS2 decreases in intensity, while the peak related to the CS1 increases, so that in the eighth step the two structures become similar in intensity and from step 11 on, the graph shows a unique structure. This implies that most of the long and mostly not branched chains have undergone degradation and are less able to form CS2 although these chains are still able to form CS1 structures.

The single peak observed is wide in the range of melting temperatures. This means that the breaking of the chains produces new intermediate substructures that implying a wider peak.

#### 3.1.3. Crystallinity

The amount of crystallinity was analyzed from the films obtained by extrusion blowing in every processing step (named film) and from the pellets obtained after the recycling of these films in every step (named pellets). Figure 6 provides further insight in the structural changes that take place during the successive processing of the samples.

The line related to the percent of crystallinity of the structure type II CS2 found in the film samples decreases with the processing steps, from a value around 10% to 5%. This indicates that with increasing of processing steps the polymeric chains are more degraded, they are shorter and more branched due to the transfer chain reactions that take place in the polymer, being increasingly difficult create structures CS2 which are related to the higher melting temperature [27].

The crystallinity type I in the film samples (CS1) has a higher percent than the CS2 due to the characteristics of the initial polymer. It follows also a slightly decreasing trend which is less marked than in CS2. The crystalline content of CS1 is also affected by the degradation, but, being related to the more branched parts of the polymer, is less affected and the decrease is only from 25% to 21% [28].

It is the most branched structure and the least ordered, so it has more possible reaction points. This means that, as the degradation steps increase, part of this crystalline structure degrades and becomes part of the amorphous phase.

When comparing the crystallinity of the samples obtained in recycled pellets an interesting phenomenon is observed. The amount of structure CS2 decreases with the processing samples, but the crystallinity in the structure CS1 increases. 

This result is related to the characteristics of the processing of the samples. When the sample is submitted to stress in extrusion blowing, quick cooling and degradation hinder the correct placement of the polymer chains to develop an ordered crystalline structure. However, these chains are still able to develop crystallinity when they are processed in the recycling machine. A slower cooling allows the development of new structures that are possibly favored by the presence of cross-links produced by radicals present in the samples or other related phenomena that are helping to rearrangement of the structure and therefore increase.

The addition of these competing tendencies, result in a total crystallinity in the samples investigated in recycled pellets, obtained from the recycling process that is approximately stable with the processing steps. The decrease of the CS2 structure is compensed by the increase of the structure CS1, there is no change in the total amount of crystalline regions but the sample will present in general a higher temperature of melting. 

In the case of the samples in film, instead, the total crystallinity is lower with processing steps, due that both structures decrease [29].

#### 3.1.4. Oxidation Temperature 

DSC included in Figure 3 shows the oxidation curve that appears as a change of baseline motivated by the exothermic process that occurs. This takes place at high temperature ranges, where the polymer reacts with the oxygen to undergo the oxidation process, which causes the breaking of the polyethylene covalent bonds. Drawing the tangent lines to the baseline and to the slope of the exothermic change as shown in Figure 3, the point where they intersect corresponds to the so-called oxidation induction temperature (T_ox_) or (OIT) [30].

Figure 7 shows the obtained values for these temperatures. In the first steps, the variation is minimal, due to the presence of antioxidants in the starting material, but after these additives have lost their ability to stops oxidation the temperature required to start the process decreases. 

The results obtained for the oxidation temperature show the same behavior as the same of other properties. The general trend is to decrease the oxidation temperature. From step 6, the film decreases with less incidence than the pellets.

The more degraded the material is, the lower the oxidation induction temperature, that is, the greater the ease and predisposition of the material to oxidize.

In relation to the previous theory, when the fragmentation of the chains and their subsequent branching occurs, radicals and branches with a greater number of weak points are generated. When subjected to a high temperature and an oxidizing atmosphere, they tend to oxidize more easily oxidation is induced at a lower temperature (T_ox_). 

Unlike the melting temperature, which depends on the secondary interaction forces, the oxidation temperature depends on the chain length and the crosslinking, which affect the C-C bonds of both, the amorphous and crystalline structures. Film-type samples are manufactured by extrusion (like pellets) and then by blowing, which facilitates the formation of more crosslinking due to the easier and better targeting of free radicals generated in degradation by previous extrusion. This issue increases meaningfully the oxidation temperature as the polymer becomes more degraded. This is the reason why in the odd steps (film samples) the oxidation temperature is higher than in even steps [31].

### 3.2. Structural Analysis FT-IR 

Figure 8 show the FTIR spectra in the range were the crystallinity of the LLDPE could be monitored. As appears in the spectra, the intensities of the bands related to the amorphous phase of LLDPE (1472 and 1462 cm^−1^) have different values. The spectra in Figure 8a correspond to the film samples (odd steps) and the spectra in Figure 8b correspond to the recycled pellets (even steps). There is clearly an important difference between both types of sample. The recycled pellet samples keep practically the same peak intensities, while the intensities of the film-type samples are very different from each other showing an increase in the band at 1472 and a decrease at 1462 cm^−1^ [27,32].

The values obtained by FTIR corroborate the total crystallinity degree measured by DSC, where the recycled pellet samples show an approximately constant crystallinity value of 33.2% (Figure 6) and then the FTIR spectra are quite similar in all samples. Instead, the crystallinity values obtained by DSC for the film-type samples, progressively decrease from 33.7% to 26.1% with a slope of 7.2%. The decrease of crystallinity values obtained previously by DSC corresponds to the increase of the amorphous phase obtained by FTIR using the 1472/1462 band ratio.

The evolution as a function of the degradation steps shows an absorbance ratio 1472/1462 cm^−1^ (related to amorphous content) of recycled pellets remains in a constant value, while in case of the film, this absorbance ratio increases significantly. The explanation of this different behavior of recycled pellets and films samples is related to the different degradation process that happens in films-type samples and recycled pellet samples. In the film samples, a breaking of the main polymeric chain is the predominant phenomenon, due to the stretching process to which the film is submitted. As mentioned previously, the chains become shorter, but these chains are still able to develop crystallinity when they are processed in the recycling machine. 

On the other hand, the samples of recycled pellets are not being subjected to the stretching while they are processed, and a cross-linked structure induced by the presence of radicals helps to keep a certain order in the structures, in a way that maintain an apparent degree of crystallinity in CS1.

The evolution of the CH_3_ group (assigned to the band of 1380–1375 cm^−1^) has also been analyzed. No changes have been observed. This may be due to the fact that in the degradation processes the creation of new CH_3_ groups that are generated in the chain lock are offset by the formation of crosslinks taking advantage of the short ramifications of the LLDPE chains.

### 3.3. Tear Resistance 

To analyze the incidence of degradation of film-type samples in the tear strength, which is going to be key in the marketplace, a normalized test has been performed. 

The results of Figure 9 show that the tear strength in longitudinal direction is lower than the transversal. The difference is caused by the orientation of the chains along the longitudinal direction, being easier to break the secondary forces that link them, compared to the transversal direction where tearing involves an increased amount of covalent bonds. Also, the preferential orientation of lamella in the spherulitic structure along transverse direction allows a higher tear value.

When the samples are submitted to the degradation process caused by the recycling, the tear strength value in longitudinal way decreases by approximately 40% for each film (every two processing steps). Meanwhile, in the case of tearing in the transverse direction, something similar occurs, the values decrease, but with less slope, approximately 20% between the different film steps. This means that the thermal degradation of the films affects more the tear strength in the longitudinal direction, because the fraction of CS1 structure slowly decreases as commented above. This leads an increase in the ratio of the row-nucleated structure, which means a greater orientation in the longitudinal direction. The anisotropy of the material is increased, producing higher differences in the tear ratio between longitudinal and transversal directions (tear LD/tear TD), which varies from a 0.66 in the first step to 0.11 to the eleventh step [17]. As noted above this is related to the arrangement of the chains with their branches and crosslinking.

## 4. Conclusions

LLDPE presents two different crystalline structures (noted as CS1 and CS2). Both structures progress differently through the recycling steps to which the samples are subjected. The structural behavior of the recycled pellet-type samples (even steps), and the recycled film-type samples (odd steps) is different. A similar decrease in CS2 is observed for both samples. On the contrary, the CS1 of the recycled pellet samples increases while the CS1 of the film samples decreases. This is because the film is submitted to the cooling stress in extrusion blowing and the degradation hinders the correct placement of the polymeric chains to develop a proper structure. However, these chains are still able to develop crystallinity when they are processed in the recycling machine, then developing new structures that are favored by the presence of crosslinking produced by radicals present in the pellets that are helping to the rearrangement of the structure. Further research on the phenomena taking place in the recycling machine and the formation of pellets is required. 

The results obtained for the oxidation temperature show a decrease of this, although from step 6, the film decreases with less incidence than the pellets. This behavior is due to manufacturing process of films (extrusion and blowing), which facilitates the formation of more crosslinking due to the easier and better targeting of free radicals generated in degradation by previous extrusion.

A very interesting correlation between the changes of crystallinity obtained by FTIR and DSC is observed, suggesting an important insight in the microstructure of the material.

The tear strength decreases through the recycling steps, especially in the longitudinal direction (40% for each film) and also in the transverse direction (20%). The results show that, up to 4 recycling cycles, the samples present properties that are adequate to their intended application, allowing the use of recycling waste in consumer bags. However, a lower tear strength could lead to further spreading of small pieces of PE in the environment.

## Figures and Tables

**Figure 1 polymers-14-00754-f001:**
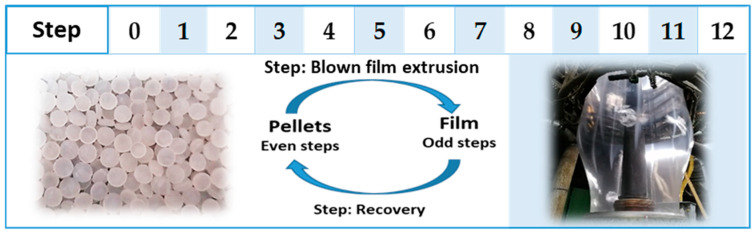
Coding of studied samples: Step zero (original LLDPE); Odd number steps (extrusion and blowing process to obtain a film); Even number steps (recover LLDPE in pellets from film).

**Figure 2 polymers-14-00754-f002:**
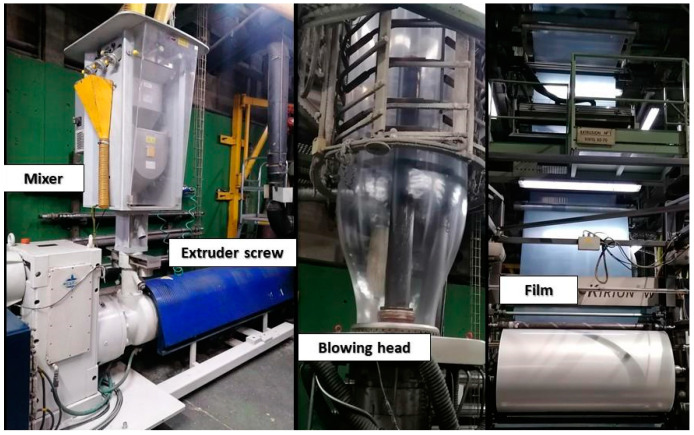
Parts of the extruder used in the study: Mixer, extruder screw, blowing head and obtained film.

**Figure 3 polymers-14-00754-f003:**
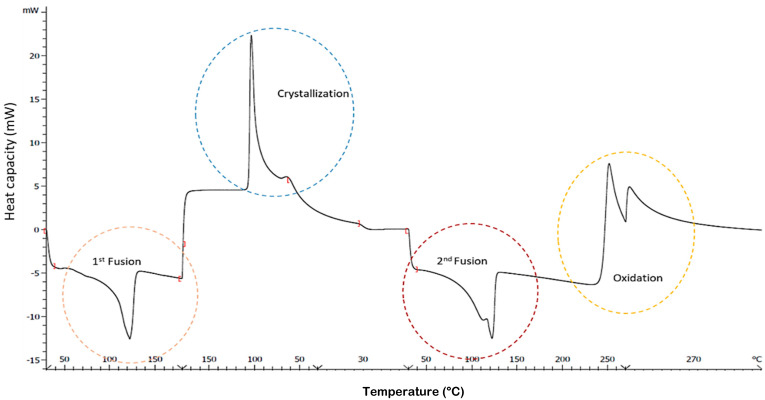
Thermogram where 1st fusion (erase of thermal history); crystallization; 2nd fusion and oxidation are observed.

**Figure 4 polymers-14-00754-f004:**
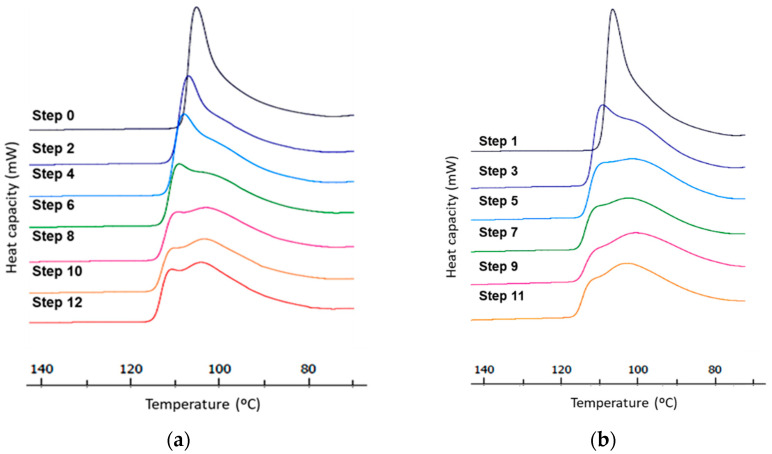
DSC crystallization thermograms for LLDPE as a function of processing steps (**a**) Steps 0, 2, 4, 6, 8, 12 correspond to recycled pellets samples obtained from the film transformed; (**b**) Steps 1, 3, 5, 7, 9, 11 to the film samples obtained from the recycled.

**Figure 5 polymers-14-00754-f005:**
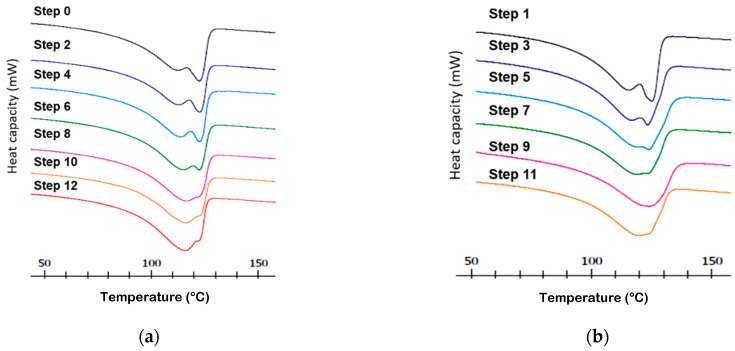
DSC fusion thermograms for LLDPE as a function of processing steps (**a**) Steps 0, 2, 4, 6, 8, 12 correspond to recycled pellets samples obtained from the film transformed; (**b**) Steps 1, 3, 5, 7, 9, 11 to the film samples obtained from the recycled.

**Figure 6 polymers-14-00754-f006:**
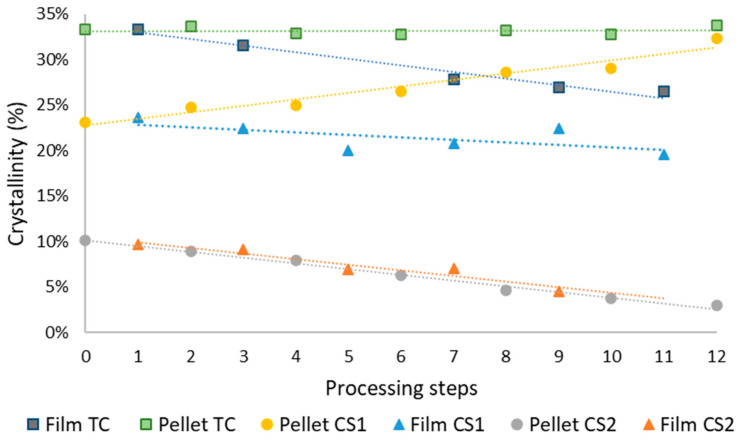
Evolution of degree of crystallinity (*χ_c_*) for each crystalline structure (CS1 and CS2) and the total amount of crystallinity with the processing steps (TC). The original data are obtained from the melting enthalpy of the samples.

**Figure 7 polymers-14-00754-f007:**
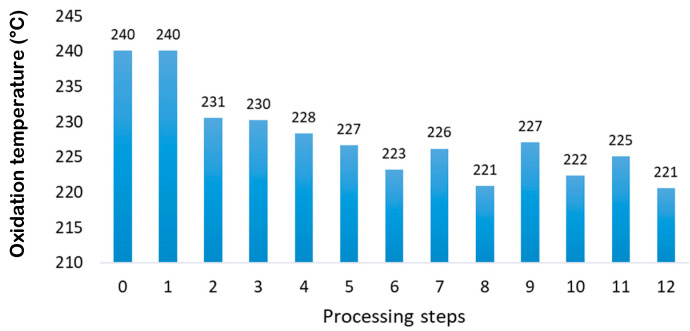
Oxidation induction temperature (OIT) as a function of the processing steps.

**Figure 8 polymers-14-00754-f008:**
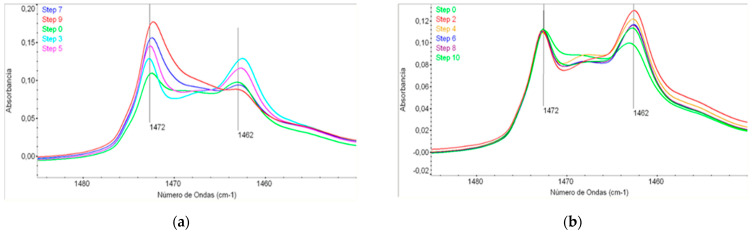
FTIR correspond to the film samples (**a**) even steps *film sample*, (**b**) Odd steps *recycled pellet*.

**Figure 9 polymers-14-00754-f009:**
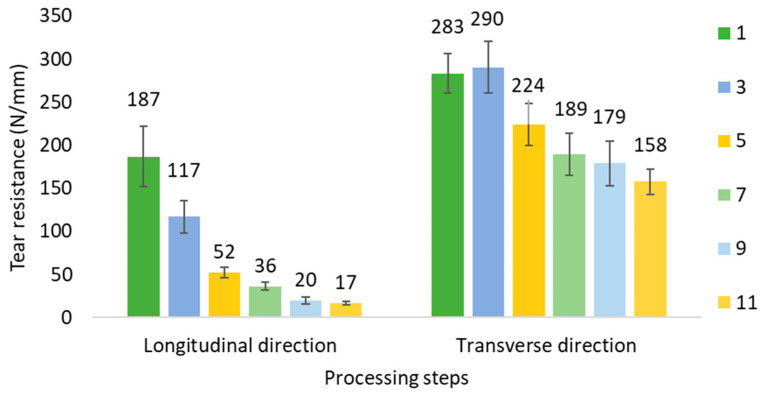
Tear strength of film samples (Longitudinal and transverse direction).

**Table 1 polymers-14-00754-t001:** DSC crystallization and fusion temperatures as function of processing steps, recycled pellet and film for structure CS1 and CS2.

Sample	Processing Step	Tc (CS1)	Tc (CS2)	Tm (CS1)	Tm (CS2)
Pellets	0	-	104.33	111.35	121.34
	2	-	106.20	111.81	121.20
	4	101.62	106.67	113.41	122.05
	6	103.10	107.48	114.72	122.22
	8	101.73	108.34	115.34	120.75
	10	102.44	109.37	115.11	120.95
	12	102.52	109.35	116.44	121.33
Film	1	-	104.89	111.14	121.76
	3	101.05	107.05	113.70	120.34
	5	99.95	106.84	115.22	121.49
	7	100.14	106.60	116.32	120.32
	9	99.73	107.40	117.73	122.09
	11	100.44	108.38	118.10	121.31

## Data Availability

Not applicable.

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
