# Peer review of "Circular Economy Assessment in Recycling of LLDPE Bags According to European Resolution, Thermal and Structural Characterization"

_polymers, 2022, doi:10.3390/polym14040754_

Round 1

Reviewer 1 Report

Dear Sir,

I understand that the idea behind this study was to test the properties of a plastic material that respect the requirement of including at least 70% of recycled material, a target that was set by the EU, as presented at the beginning of the abstract. That “Circular Economy Package” mentioned by the authors should be given a reference and some consideration must be made on how the value of 70% was set – I assume that the European authorities did not jump on this value on a hunch and that some tests were carried out previously to setting this value.

Anyway, the idea is very interesting, but studies in that aspect were already carried out, but in a somehow different way. Please, for LLDPE compare with the study of Jin et al. (The effect of extensive mechanical recycling on the properties of low density polyethylene, Polymer Degradation and Stability, 2012, 97(11), 2262-2272) or Saikrishnan et al. (ref. 15 of the manuscript) and mention also similar studies carried out for high-density PE (e.g. Oblak et al., Processability and mechanical properties of extensively recycled high density polyethylene, Polymer Degradation and Stability, 2015, 114, 133-145) or Cruz and Zanin (Evaluation and identification of degradative processes in post-consumer recycled high-density polyethylene, Polymer Degradation and Stability, 2003, 80(1), 31-37). I suggest to look also toward HDPE because the processes that occurs during the cycles of film extrusion – pellets recovery may cause a widening of the polymerization degree simultaneously with a growing branching/reticulation process. The polymerization degree may become lower, but also higher, and therefore, cumulated with a better reticulation, lead the LLDPE toward HDPE. Comparing with results obtained for HDPE may allow to confirm or infirm the fact that the degree of polymerization decreases along with the growing number of cycles.

For the degradation patterns of LLDPE, refer to the works of Al-Salem et al. (Journal of Material Cycles and Waste Management, 2019, 21, 1106-1122; Journal of Polymer Research, 2018, 25. art. nr. 111).

Does the crystalline structures differ much from those obtained by Craig and White (Crystallization and chemi-crystallization of recycled photodegraded polyethylenes, Polymer Engineering and Science, 2005, 25(4), 588-595)?

The recorded oxidation temperatures presents an interesting pattern: after the first 5-6 steps, the OIT of pellets seems to be always lower than that of the films. Why would that be? According to Anderson et al. (Degradation of polyethylene during extrusion. II. Degradation of low-density polyethylene, linear low-density polyethylene, and high-density polyethylene in film extrusion, Applied polymer, 2004, 91(3), 1525-1537) oxidative degradation occur during extrusion. But what about the phenomenon that occur during pellet reformation?

IR analysis: the authors followed mainly the evolution of the absorption bands at 1472 and 1462cm−1, which are the scissoring and stretching of the CH2 and CH3 moieties. I suggest to look up also on the 2914 and 2847 cm−1 (or nearby) (asymmetric and symmetric stretching of the CH2 and CH3 moieties), but also at ca. 718 cm−1 (CH2 rocking). But the most interesting result is the variation of the ratio 1472/1462 peaks. Since 1462 peak corresponds to the CH3 groups and 1472 to the CH2 groups, the evolution of these peaks confirm that during extrusion the number of CH3 groups is augmenting (fig. 8b), confirming numerous chain breaking, but also branching. The question that remains, is what is happening during pellet formation that seems to reverse the process? More interesting, in Fig. 8a, as the number of cycles in growing, the ration CH2 / CH3 is also augmenting, suggesting longer chains formation (more CH2, less CH3).

The tear resistance test is not very promising, since it shows that recycling can be made only once (1 being the first film, with the original pellets and 3 being the film obtained after extrusion-recovery-extrusion). In that aspect, the use of the term “cycle” could be detrimental, since the sentence in the Conclusion paragraph: “The results show that, up to 4 recycling cycles, the samples present properties that are adequate to their intended application, allowing the use of recycling waste in consumer bags.” could be interpreted as LLDPE in plastic bags could be recycled up to 4 times, while in fact only one recycling process would be feasible, after the second one the resistance becoming much weaker.

I suggest to add in the Conclusion paragraph a sentence asking for the investigation of the pellet recovery process in order to improve the properties of the pellets and obtaining through extrusion LLDPE films with similar properties, even after a higher number of cycles. In the same time, low resistance to tearing could increase the danger of LLDPE toward the environment through easier formation of microplastics.

Minor comments:

- there some minor grammar and synthax errors that must be addressed – I will give only a few examples, but the entire manuscript must be re-checked: Abstract: “The process is analogue to the industrial, starts from samples in pellets …” – change to: “The process is analogue to the industrial, it starts from samples in pellets …”; “… the bands relation suggested by [9] was used …” – change to “… the bands relation suggested by Zerbi et al. [9] was used …”

- Abstract: “…. but both show a clear tendency to decrease due to oxidation, …” point out exactly what is decreasing (I assume it is the degree of polymerization)

- I wouldn’t use the term “cycle”, because a cycle means coming back at the same point were the process starts, which would mean the pellets: thus 1+2 makes a cycle. I suggest to use the term “steps”, 2 consecutive steps making a cycle. Thus, fig. 1 would have to be modified as follows: pellets and films remain, but under them odd and even cycles disappear. Extrusion and recovery remains, but under them even or odd step appears. Samples will therefore designated by numbers: “0” is the initial pellets, odd numbers will be films obtained after odd steps (extrusions) and even numbers will be pellets obtained after even steps (recoveries).

- remain consistent with the units (ex. for flow keep either “kg/h” or “kg/hour”);

Overall, the manuscript is very interesting and is worth publication, but nevertheless it needs some revision before full acceptance.

Author Response

I understand that the idea behind this study was to test the properties of a plastic material that respect the requirement of including at least 70% of recycled material, a target that was set by the EU, as presented at the beginning of the abstract. That “Circular Economy Package” mentioned by the authors should be given a reference and some consideration must be made on how the value of 70% was set – I assume that the European authorities did not jump on this value on a hunch and that some tests were carried out previously to setting this value.

According to the proposal of the reviewer we have included a reference to the Directive 94/62/EC on packaging and packaging waste:

Directive 94/62/EC on packaging and packaging waste: https://eur-lex.europa.eu/legal-content/EN/TXT/?uri=celex%3A31994L0062 access 11/01/2022

The reviewer also points out a very interesting question: how the European authorities set the values defined as targets for the next years. The complicated mechanism of decision making in the European union is beyond the authors’ area of expertise. The information available is that these directives are based on different reports. The quality of each report is checked by an independent body, the Regulatory Scrutiny Board, which issues opinions. Trying to find the authors of the reports or boards only led us to designations as “European directory” or similar general terms.

Anyway, the idea is very interesting, but studies in that aspect were already carried out, but in a somehow different way. Please, for LLDPE compare with the study of Jin et al. (The effect of extensive mechanical recycling on the properties of low density polyethylene, Polymer Degradation and Stability, 2012, 97(11), 2262-2272) or Saikrishnan et al. (ref. 15 of the manuscript) and mention also similar studies carried out for high-density PE (e.g. Oblak et al., Processability and mechanical properties of extensively recycled high density polyethylene, Polymer Degradation and Stability, 2015, 114, 133-145) or Cruz and Zanin (Evaluation and identification of degradative processes in post-consumer recycled high-density polyethylene, Polymer Degradation and Stability, 2003, 80(1), 31-37). I suggest to look also toward HDPE because the processes that occurs during the cycles of film extrusion – pellets recovery may cause a widening of the polymerization degree simultaneously with a growing branching/reticulation process. The polymerization degree may become lower, but also higher, and therefore, cumulated with a better reticulation, lead the LLDPE toward HDPE. Comparing with results obtained for HDPE may allow to confirm or infirm the fact that the degree of polymerization decreases along with the growing number of cycles.

We have followed the recommendations of the reviewer and have included these references and some comments in the revised text.

The interesting study of Jing et al. focused in LDPE, their results showed that, when LDPE is extruded several times too basic processes related to degradation occur. There is a scission of chains, which is a common polymer degradation phenomenon, and crosslinking. These results are in consonance with the ones presented in the manuscript. During the discussion we have related the differences in crystallization and crystallinity to these two processes. While the scission tends to decrease thermal parameters of the polymer, crosslinking tends to the opposite. Is the balance between these two processes which causes the different behaviors that we find in the studied samples. The changes in melting temperature, which are not deeply discussed in Jing et al are not very different to the presented in this paper, but observing the whole DSC melting curve is more informative and provides more insight in the phenomena that are taking place in the polymer.

Saikrishnan et al explain the degradation mechanisms, show the decrease of viscosity caused by the recycling and also the decrease in properties. Like in the present manuscript, although there is a decrease in properties caused by the recycling authors still consider the recycled material able to be applied in consumer products.

Oblak et al propose a predominance of the branching of PE in the first cycles of extrusion and after that an increase of the scission of the chains. Regarding these results, some other aspects may have relevance. The combination of a specific polymer, the conditions of extrusion, temperature, speed, may have an effect in the predominance of a phenomenon or the other. Oblak et al themselves report that some researchers have detected an increase in MFI with extrusion Cycles while they find, instead, a decrease. At the same time, the effect of branching in some measurements can be very similar to the crosslink (i. e. viscosity, MFI, torque.. ). They also find a general decrease in crystallinity but the do not discuss the presence of different crystalline structures a DSC graphic. An important consideration also discussed below is that our processing includes blow film, (even if there is a reprocessing to pellets) for this reason we have cited the work of Zhang et al who refer also to blow film.

Results of Cruz and Zanin follow also the logic of the present manuscript showing oxidation of the reprocessed material and a decrease of MFI, suggesting crosslink.

According to the reviewer suggestion we have improved our discussion of the results. The bibliography cited above is significative of how the different processes involved, oxidation, branching, crosslinking, scission (related to molecular weight variations) affect simultaneously to the behavior of the material and the predominance of one or another is relevant also in the microstructural arrangements of the molecular chains.

For the degradation patterns of LLDPE, refer to the works of Al-Salem et al. (Journal of Material Cycles and Waste Management, 2019, 21, 1106-1122; Journal of Polymer Research, 2018, 25. art. nr. 111).

We have included the reference and citation in the manuscript

Does the crystalline structures differ much from those obtained by Craig and White (Crystallization and chemi-crystallization of recycled photodegraded polyethylenes, Polymer Engineering and Science, 2005, 25(4), 588-595)?

In their interesting work Craig and White only appreciated (in all polymers studied) a type of crystalline structure, that could be related to one of the crystalline structures observed I this study. However, the crystalline structures that we took as references for our study were those obtained in a process of blow film (reference Zhang et al) because of the importance of the processing in the developing of polymeric microstructures. Anyway, it is also interesting that Craig and White found recrystallization phenomena in their samples, which are submitted to theoretically less drastic conditions than ours.

The recorded oxidation temperatures presents an interesting pattern: after the first 5-6 steps, the OIT of pellets seems to be always lower than that of the films. Why would that be? According to Anderson et al. (Degradation of polyethylene during extrusion. II. Degradation of low-density polyethylene, linear low-density polyethylene, and high-density polyethylene in film extrusion, Applied polymer, 2004, 91(3), 1525-1537) oxidative degradation occur during extrusion. But what about the phenomenon that occur during pellet reformation?

Unlike the melting temperature, which depends on the secondary interaction forces, the thermal decomposition temperature depends on the chain length and the crosslinking, which affect the C-C bonds of both, the amorphous and crystalline structures. Film-type samples are manufactured by extrusion (like pellets) and then by blowing, which facilitates the formation of more crosslinking due to the easy better targeting of free radicals generated in degradation by previous extrusion. This issue increases meaningfully the temperature of thermal decomposition as the polymer becomes more degraded.

IR analysis: the authors followed mainly the evolution of the absorption bands at 1472 and 1462cm−1, which are the scissoring and stretching of the CH2 and CH3 moieties. I suggest to look up also on the 2914 and 2847 cm−1 (or nearby) (asymmetric and symmetric stretching of the CH2 and CH3 moieties), but also at ca. 718 cm−1 (CH2 rocking). But the most interesting result is the variation of the ratio 1472/1462 peaks. Since 1462 peak corresponds to the CH3 groups and 1472 to the CH2 groups, the evolution of these peaks confirm that during extrusion the number of CH3 groups is augmenting (fig. 8b), confirming numerous chain breaking, but also branching. The question that remains, is what is happening during pellet formation that seems to reverse the process? More interesting, in Fig. 8a, as the number of cycles in growing, the ration CH2 / CH3 is also augmenting, suggesting longer chains formation (more CH2, less CH3).

The 1472/1462 doublet used in this manuscript was proposed by Zerbi et al. and used for other referenced manuscripts to determine the crystalline phase content of PE, where the 1472 band is related to CH2 bending vibration of the crystalline phase, and the band of 1462 is related to bending vibrations of the CH2 of the amorphous phase.  In this manuscript this relation of bands has been used in order to corroborate the results obtained by means of DSC. On the other hand, as the referee rightly points out, to follow the evolution of the methyl group (CH3), the band of 1380 -1375 cm-1 must be analyzed. According to attached picture the spectral band at 1377 cm-1 of both groups of samples almost does not change. This may be due to the fact that in the degradation processes the creation of new CH3 groups that are generated in the chain lock are offset by the formation of crosslinks taking advantage of the short ramifications of the LLDPE chains.

On the other hand, the bands at 2914 and 2847 cm− 1 (or nearby) (asymmetric and symmetric stretching of the CH2 and CH3 moieties), and doublet at 730-718 cm− 1 (CH2 rocking), do not provide any additional information regarding the evolution of crystallinity.

We have improved the discussion of these results according to the suggestions of the reviewer

 The tear resistance test is not very promising, since it shows that recycling can be made only once (1 being the first film, with the original pellets and 3 being the film obtained after extrusion-recovery-extrusion). In that aspect, the use of the term “cycle” could be detrimental, since the sentence in the Conclusion paragraph: “The results show that, up to 4 recycling cycles, the samples present properties that are adequate to their intended application, allowing the use of recycling waste in consumer bags.” could be interpreted as LLDPE in plastic bags could be recycled up to 4 times, while in fact only one recycling process would be feasible, after the second one the resistance becoming much weaker.

The appreciation of the review is right but, according to the company promoting the studio the properties of the recycled films are enough to be used in their products.

We have introduced the changes proposed by the reviewer, the word Cycle is certainly not appropriated and we have substituted it by “step”. We have also rephrased the paragraph.

I suggest to add in the Conclusion paragraph a sentence asking for the investigation of the pellet recovery process in order to improve the properties of the pellets and obtaining through extrusion LLDPE films with similar properties, even after a higher number of cycles. In the same time, low resistance to tearing could increase the danger of LLDPE toward the environment through easier formation of microplastics.

 Following this recommendation, we have modified the paragraph

Minor comments:

- there some minor grammar and synthax errors that must be addressed – I will give only a few examples, but the entire manuscript must be re-checked: Abstract: “The process is analogue to the industrial, starts from samples in pellets …” – change to: “The process is analogue to the industrial, it starts from samples in pellets …”; “… the bands relation suggested by [9] was used …” – change to “… the bands relation suggested by Zerbi et al. [9] was used …”

- Abstract: “…. but both show a clear tendency to decrease due to oxidation, …” point out exactly what is decreasing (I assume it is the degree of polymerization)

- I wouldn’t use the term “cycle”, because a cycle means coming back at the same point were the process starts, which would mean the pellets: thus 1+2 makes a cycle. I suggest to use the term “steps”, 2 consecutive steps making a cycle. Thus, fig. 1 would have to be modified as follows: pellets and films remain, but under them odd and even cycles disappear. Extrusion and recovery remains, but under them even or odd step appears. Samples will therefore designated by numbers: “0” is the initial pellets, odd numbers will be films obtained after odd steps (extrusions) and even numbers will be pellets obtained after even steps (recoveries).

- remain consistent with the units (ex. for flow keep either “kg/h” or “kg/hour”);

 We have made all the changes recommended by the reviewer, which are very useful in order to improve the manuscript

Overall, the manuscript is very interesting and is worth publication, but nevertheless it needs some revision before full acceptance.

Reviewer 2 Report

The authors study the recycling of the linear-low density polyethylene plastic. It is an important study to understand the tear strength of the plastic after several recycling activities. However, there are mistakes found in the articles and I suggest the following revisions to be implemented in the manuscript.   

  1. It would be better for the author to state the key findings from the research work in the abstract section and quantify it.

  1. Avoid using “we”, “our” in writing the article.

  1. The objective of the study is unclear. It would be better for the author to highlight the state of the art for the current study and the gap from the previous literature.

  1. Label for figure 8 is too small including axis label.

  1. It would be better for the author to quantify the tear strength in in the conclusion section.

Author Response

The authors study the recycling of the linear-low density polyethylene plastic. It is an important study to understand the tear strength of the plastic after several recycling activities. However, there are mistakes found in the articles and I suggest the following revisions to be implemented in the manuscript.  

It would be better for the author to state the key findings from the research work in the abstract section and quantify it.

 We think the recommendation of the reviewer constitutes an improvement of the manuscript and  we have modified the paragraph accordingly

Avoid using “we”, “our” in writing the article.

We agree, following this recommendation, we have modified the manuscript

The objective of the study is unclear. It would be better for the author to highlight the state of the art for the current study and the gap from the previous literature.

According to this suggestion we have modified the introduction and abstract.

Label for figure 8 is too small including axis label.

We have increased the size of the label

It would be better for the author to quantify the tear strength in in the conclusion section.

We agree and we have modified the section accordingly

Round 2

Reviewer 2 Report

accepted as it is

Author Response

Thank you very much for accepting our manuscript.
